# A Comparison of Clinical Features of Youth with and without Rhinitis Signs and Symptoms Who Are Hospitalized for Headache

**DOI:** 10.3390/children9081241

**Published:** 2022-08-17

**Authors:** Si-Jia Tang, Heejin Lee, Tiantian Cui, Jae Min Lee, Ji Young Ahn, Sua Lee, Saeyoon Kim

**Affiliations:** 1Graduate School, Yeungnam University College of Medicine, Daegu 42415, Korea; 2Department of Pediatrics, Yeungnam University Medical Center, Daegu 42415, Korea; 3Department of Pediatrics, Yeungnam University College of Medicine, Daegu 42415, Korea

**Keywords:** headache, allergic rhinitis, migraine, children

## Abstract

Headache and allergic rhinitis (AR) are common in children and often co-occur. We investigated the clinical characteristics of pediatric headaches and the association of AR and chronic headaches. We retrospectively reviewed the medical records of patients admitted to our pediatric inpatient and outpatient clinics with complaints of headache between January 2017 and June 2020 for headache-specific history, AR signs and symptoms, allergy skin prick test, inhalant multiple allergen simultaneous test results, laboratory and imaging findings, and medication history. The patients were divided into three subgroups: AR, non-AR, and headache groups, reporting 45.7% patients with headache alone, 13.7% with additional AR, and 31.6% with abnormal imaging findings, suggesting that headache was combined with sinusitis (24.3%) or mastoiditis (7.3%). Furthermore, 6% of the patients had both AR and sinusitis. Body mass index (BMI) differed significantly between the AR and the non-AR and headache groups (*p* = 0.03). The BMI differed significantly according to headache severity (*p* ˂ 0.001). The most common allergen was “dust or mites” (41.1%). Acetaminophen (35.9%) was the most commonly used painkiller. The coexistence of AR and headache may indicate that these conditions share a similar pathophysiology. Better management of allergies may facilitate diagnosis, treatment, and prophylaxis of headaches.

## 1. Introduction

Headaches are common among children and adolescents, and they can persist into adulthood [1]. The International Headache Society (IHS) classifies the majority of pediatric headaches as either primary (e.g., migraine, tension-type headache, cluster headache, and many less common primary headache disorders) or secondary; the latter headaches are “attributed to” other disorders, such as upper airway infection, influenza, sinusitis, head trauma, or intracranial disease [2]. Migraine headaches have a negative impact on daily life by impairing academic performance, decreasing family interactions, and interfering with social activities [3]. Similarly prevalent in children, allergic rhinitis (AR) is frequently associated with other comorbidities, such as asthma, middle ear effusion, sinusitis, and headache [4]. Histamine and nitric oxide (NO) are both important inflammatory mediators of AR and migraine [5]. Sinusitis is also a common cause of secondary headaches in children, and allergic symptoms may manifest as sinusitis [6]. Several studies have described the relationship between migraine and AR, including pathophysiological resemblance, clinical similarities, and common triggers [7,8,9]. However, most of these studies have targeted adults.

Numerous studies have examined the association between migraine and allergies, although the correlation between these two conditions remains controversial [10]. We analyzed the clinical characteristics of pediatric headaches and the association of other comorbidities( such as AR, which is one of the most common allergic diseases in children) on chronic headaches [11].

## 2. Materials and Methods

### 2.1. Cases and Data Collection

This retrospective study was conducted at the Department of Pediatrics of Yeungnam University Hospital in Daegu, Korea. Between January 2017 and June 2020, 412 patients were admitted with headache as their presenting symptom.

In this study, patients were enrolled according to their headache symptoms. Therefore, not only primary headache such as migraine but all headaches were included. Chronic headaches were defined as headaches longer than 3 months according to the international classification of headache disorders third edition [12]. Their medical records were reviewed for clinical symptoms, headache severity (numerical rating scale), evidence of AR (nasal signs and symptoms, allergy test (skin prick or inhalant multiple allergen simultaneous test) results), laboratory and imaging data, and medication history.

Figure 1 depicts the inclusion/exclusion process of the patients. A total of 234 patients with headache, who underwent imaging, were included. Patients under 5 years of age (who were less likely to have allergic diseases) and those over 18 years of age were excluded. Other exclusion criteria included neurological conditions that might contribute to headaches (stroke, cerebral palsy, trigeminal neuralgia, or a seizure disorder) or a known intracranial lesion. A total of 234 patients were included in the study and divided into three subgroups: (i) allergic rhinitis (AR) Group (32 patients with positive allergy tests and AR signs and symptoms); (ii) Non-AR Group (61 patients with rhinitis signs and symptoms, without allergy tests); and (iii) Headache only Group (141 patients without rhinitis signs and symptoms). We performed a detailed chart review, including:Medical records
AgeSexBody Mass Index (BMI)Headache-specific history: Time of onset, duration, and severity (numerical rating scale)Associated allergic disease: Atopic dermatitis, asthma, allergic conjunctivitisSymptoms of rhinitis: Nasal congestion, sneezing, itching, rhinorrheaFamily history (headache, allergic disease)Medical background (headache and allergy medications)Treatment and clinical outcomesExamination data
Laboratory results: White blood cell (WBC) count, hemoglobin (Hb), platelet (PLT), aspartate aminotransferase (AST), alanine aminotransferase (ALT), blood urea nitrogen (BUN), creatinine (Cre), sodium (Na), potassium (K), and total Immunoglobulin E (IgE) levels.Allergy test: skin prick test (SPT), inhalant multiple allergen simultaneous test (inhalant MAST), and screening for common allergens (animal dander, dust, mites, mold, and pollen).Imaging data: Brain magnetic resonance imaging (MRI), brain computed tomography (CT), and X-ray paranasal sinus (PNS) series and water view.


### 2.2. Statistical Analysis

All statistical analyses were performed using IBM SPSS ver. 25.0 software (IBM Co., Armonk, NY, USA). Quantitative data were compared using ANOVA. The Kolmogorov–Smirnov test was used to confirm that the data follow normality, and a parametric statistical method was used. Correlations among the groups were determined using Fisher’s exact test. Values are presented as means ± standard deviations, and a *p*-value < 0.05 was considered statistically significant.

## 3. Results

Of the initial 412 patients, 234 met the inclusion criteria and classified as follow: AR group 13.7%, Non-AR group 26.1%, and Headache-only group 60.1%. Among them, 117 were male and 117 were female. The sex ratio did not differ among the groups (*p* = 0.831). Furthermore, 31.6% of all patients had sinusitis, and 6% had both AR and sinusitis. The mean age of the study population was 11.37 ± 3.52 years; there was no significant difference among the groups (*p* = 0.861). The BMI differed significantly between the groups (*p* = 0.03). In the AR group, the BMI was 22.19 ± 5.11 kg/m^2^, which was greater than that in the non-AR (20.41 ± 3.42 kg/m^2^) and headache groups (20.17 ± 3.76 kg/m^2^) (Table 1). We investigated basic routine laboratory tests to identify other underlying conditions that could influence the results and found that the laboratory parameters did not differ significantly among the groups (Table 2).

Of the 234 patients who underwent imaging examinations, brain MRI was performed on 185 patients, X-ray PNS or Water’s view on 48 patients, and brain CT on one patient; 31.6% of these patients yielded abnormal findings, of which 56 were on brain MRI and 18 on X-ray PNS or Water’s view. This suggested that the headache was combined with either sinusitis (*n* = 163) or mastoiditis (*n* = 17) (Table 3). Most cases of sinusitis affected the ethmoidal or maxillary sinuses (Figure 2).

Eighteen patients had positive SPT and 14 had a positive reaction to the inhalant MAST. The most frequent allergens were dust or mites (41.1%), followed by pollen (28.6%), animal dander (23.2%), and mold (7.1%) (Figure 3). We accessed pain severity by using numerical rating scale (NRS) that strongly correlated with visual rating scale [13]. NRS scores were categorized as follows: no pain; 0, mild pain; 1–3, moderate pain; 4–6, severe pain; 7–10 [14]. We found no significant relationship between headache severity and AR (*p* = 0.522) or sinusitis (*p* = 0.927). However, BMI affected the headache severity significantly (*p* < 0.001) (Table 4). Acetaminophen (35.9%) was the most commonly used medication, followed by ibuprofen and dexibuprofen (34%) (Figure 4). With the increasing severity of headache, the usage rate of combination drugs, such as phenylephrine, and prophylactic treatment drugs had also increased.

## 4. Discussion

A headache is a universal human experience. A primary headache is diagnosed when there is no obvious disease or structural problems associated with the headache. Common primary headache disorders defined by IHS include migraine and tension-type and cluster headaches. A secondary headache is one that is caused by an underlying disease. Infection, head injury, vascular disorders, brain hemorrhage, and tumors are among the different causes of secondary headache [15,16]. Additionally, sinus disease is a major cause of secondary headache [17].

AR is a prevalent and persistent childhood illness [18]. Associations between migraine and atopic disorders have been reported in both adults and children [19]. The proportion of migraine patients with a history of allergies was higher than that of non-allergic patients [20]. Both headaches and allergies impose considerable burdens on young patients and their families [21,22]. In the present study, 93 of 234 patients (39.8%) with headache had rhinitis symptoms, and 60.2% had headache only. A total of 31.6% exhibited abnormal imaging findings, which suggested that the headache was associated with sinusitis (24.3%) or mastoiditis (7.3%).

Imaging revealed that the majority of sinusitis cases involved the ethmoidal and maxillary sinuses; 6% of the patients had both AR and sinusitis. BMI significantly affected headache severity (*p* ˂ 0.001). However, no significant relationship was found between headache severity and AR (*p* = 0.522) or sinusitis (*p* = 0.927).

According to reports, the majority of patients who present with “sinus headache” have migraine [23]. The IHS diagnostic criteria for acute sinus headache include purulent nasal discharge, abnormal findings on imaging (MRI, CT, or X-ray), sinusitis, concurrent headache localized to the area of sinus disease, and disappearance of headache after treatment of acute sinusitis [8]. Of all patients, only 13% had migraine alone, 19% (of the migraine patients) also had AR, 11% had sinusitis, and 6% had both AR and sinusitis [10]. Migraine and AR share several clinical characteristics [24]. As the symptoms often overlap, migraine and headaches caused by AR are easily confused [21]. Weather changes, seasonal variations, exposure to allergens, and changes in altitude frequently trigger “sinus headache”. Grass/trees, dust, food substances, cat/dog dander, and mold are among the common allergens [9]. A study conducted between 2005 and 2013 found that 13.5% of migraine patients report seasonal exacerbations [25]. The cost of migraine treatment is higher during spring and fall, when many migraine patients with co-existing AR report increased headache intensity and frequency [26]. The correlation between migraine, allergic diseases, and nasal and sinus-related diseases requires further investigation.

AR is an IgE-mediated phenomenon involving the activation and degranulation of mast cells and basophils and often causes “sinus headache”. Histamine is a key inflammatory mediator released during degranulation, and the nasal mucosa is in close proximity to the CNS vasculature. Thus, AR and migraine share similar neural pathways [8].

Several clinical studies have shown that nasal endoscopic findings yield objective evidence of nasal congestion and rhinorrhea during migraine attacks. As is true of AR, midfacial or nasal pain during migraine may be accompanied by the release of a variety of neuropeptides, such as histamine, substance P, NO, vasoactive intestinal peptide, TNF-α, and others [15]. Many previous studies have shown that a histamine-free diet or avoidance of triggers (environmental allergens or inhalant irritants) reduces attack frequency or terminates headaches, especially in children [7]. Therefore, pediatricians should explore not only common cephalalgic triggers but also sneezing, familiarity with allergic diseases, nasal discharge, and nasal patency, in addition to consulting with an allergologist or otorhinolaryngologist if necessary [27].

The most commonly used analgesics for headache pain include aspirin, naproxen, ibuprofen, acetaminophen, combinations of these agents, triptans, and opioids [8]. In the present study, the most commonly used painkiller is acetaminophen, followed by ibuprofen, dexibuprofen. According to the increasing severity of headache, the usage rate of combination drugs, phenylephrine, and prophylactic treatment drugs also increased. Although no statistical differences were found in our study, it provides the possibility for further research. Future research can conduct more detailed grouping of headache severity and explore the correlation between these types of drugs and headache severity.

Obesity is associated with chronic daily headaches [28]. Both headache and obesity are prevalent disabling disorders associated with genetic and environmental risk factors [29]. Migraine patients with a high BMI reported more frequent episodes of headaches and higher levels of disability. Obesity is not associated with migraine onset; however, it is linked with an increased frequency and severity of chronic migraine [29]. The levels of several inflammatory mediators are elevated in obese individuals and may increase the frequency, severity, and duration of migraine attacks [30]. The link between chronic headache and obesity is worth investigating.

The most commonly used analgesics for headache pain include aspirin, naproxen, ibuprofen, acetaminophen, combinations of these agents, triptans, and opioids [8]. In the present study, the most commonly used painkiller was acetaminophen, followed by ibuprofen and dexibuprofen. With the increasing severity of headache, the usage rate of combination drugs, phenylephrine, and prophylactic treatment drugs has also increased. Although no statistical differences were found in our study, this provides a possibility for further research.

Our study has some limitations. This study was a retrospective study, and due to the different financial status and age of patients, the type and number of instrumented examinations were different. Sensitivity of the skin prick test and MAST are different, which may have affected the results [31]. Patients were reviewed retrospectively and enrolled based on symptoms rather than diagnostic criteria. Therefore, we confirmed relation of headaches and allergic rhinitis regardless of headache types since not all patients were evaluated based on migraine criteria. Furthermore, because nasal symptoms were not scored, the relationship between rhinitis symptom scores and headache severity could not be assessed. Finally, a small number of patients were enrolled because only patients who underwent imaging studies were included. The results of this study could be affected by this insufficient number of patients.

## 5. Conclusions

In our study, we found no association between headaches and AR. However, recent studies support the idea that headaches and allergic diseases are correlated [10,32]. In this study, because only patients who underwent imaging studies were included, a small number of patients may have contributed to these results. Similar pathophysiological mechanisms may be implicated rather than a close etiological association [5]. Allergic reactions may cause migraine attacks; however, the underlying mechanisms remain unclear. Although many studies have explored the relationship between headaches and AR in adults, few pediatric studies have been conducted. When encountering a migraine patient, pediatricians should consider allergic disease and its manifestations. An exploration of allergy history early in diagnosis may help guide the treatment and prophylaxis of migraine headache patients [21].

## Figures and Tables

**Figure 1 children-09-01241-f001:**
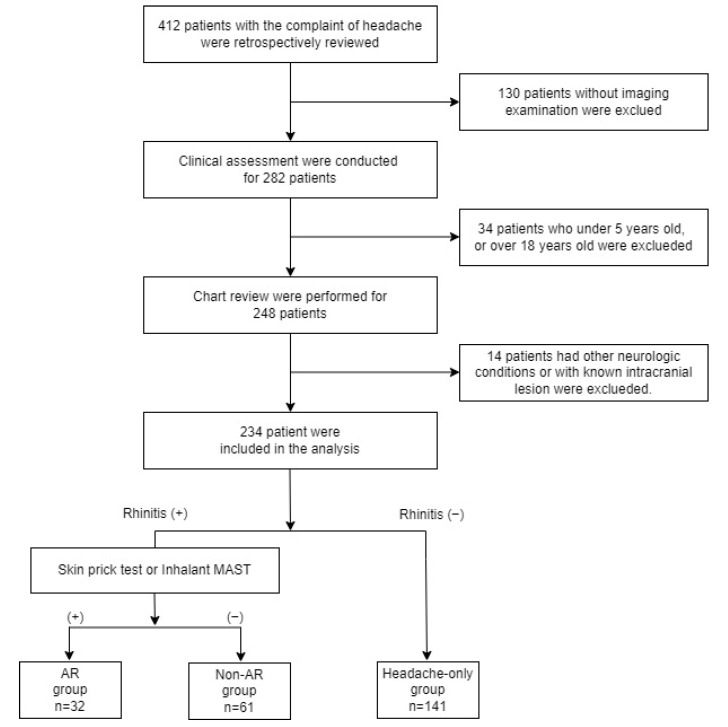
Flowchart detailing the inclusion and exclusion criteria and grouping criteria of the study subjects.

**Figure 2 children-09-01241-f002:**
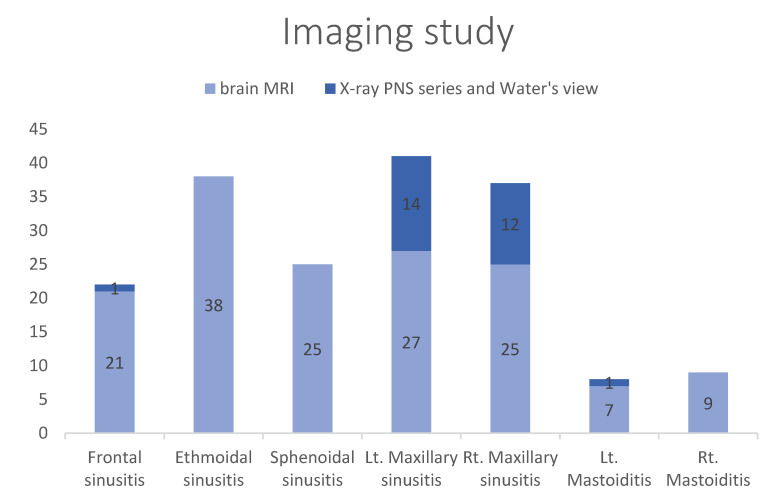
Imaging studies were performed in 234 patients. A total of 31.6% of the patients with abnormal imaging findings (56 patients following brain MRI and in 18 patients on X-ray PNS or Water’s view). Most sinusitis lay in the ethmoidal or maxillary sinuses. The *Y*-axis is the number of patients.

**Figure 3 children-09-01241-f003:**
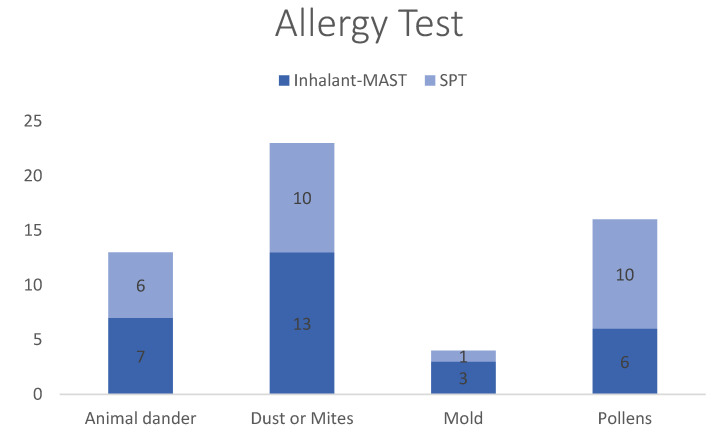
Allergy test of the patients with AR in the study revealed the most frequent allergen was “dust or mites” (*n* = 23, 41.1%) and the remaining allergens, in decreasing order, included “pollen” (*n* = 16, 28.6%), “animal dander” (*n* = 13, 23.2%), and “mold” (*n* = 4, 7.1%). The *Y*-axis is the number of patients.

**Figure 4 children-09-01241-f004:**
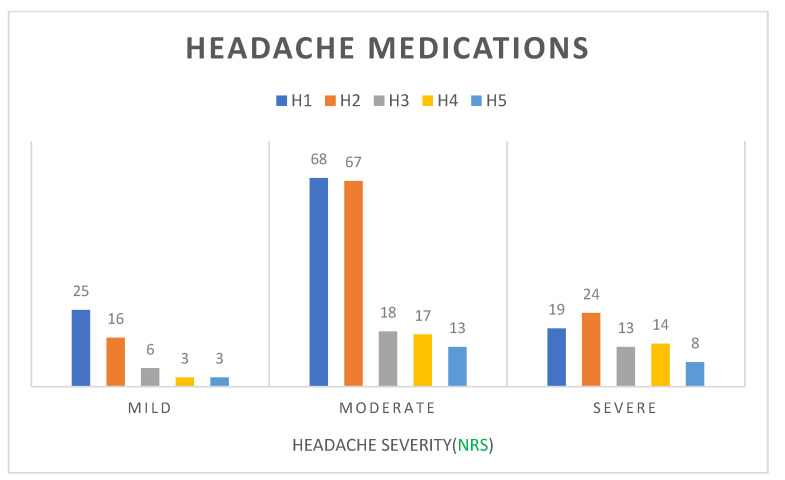
Different severity of headache (NRS) patients’ commonly used medications. The most commonly used painkiller is acetaminophen, followed by ibuprofen and dexibuprofen. According to the increasing severity of headache, the usage rate of H4 and H5 drugs also increased. The *Y*-axis is the number of patients. NRS, numerical rating scale. H1, Acetaminophen (Tylenol, Setopen); H2, Ibuprofen, Dexibuprofen (Carol-F, Anyfen); H3, Naproxen (Anaprox); H4, Combination drugs (Ultracet), Phenylephrine (Mydrin); H5, Prophylactic treatment: Flunarizine (Sibelium), Topiramate (Topamax).

**Table 1 children-09-01241-t001:** Comparison of age and BMI from the AR, non-AR, and headache-only groups.

Variable	AR Group	Non-AR Group	Headache-Only Group	*p*-Value
	*n* = 32 (13.7%)	*n* = 61 (26.1%)	*n* = 141 (60.1%)	
Age	11.38 ± 4.00	11.16 ± 3.23	11.46 ± 3.56	0.861
BMI	22.19 ± 5.11	20.41 ± 3.42	20.17 ± 3.76	0.03 *

Values are presented as mean ± standard deviation. BMI, Body Mass Index. * Statistically significant.

**Table 2 children-09-01241-t002:** Comparison of laboratory data from the allergic rhinitis (AR), non-AR, and headache groups.

Variable	AR Group	Non-AR Group	Headache Group	*p*-Value
	*n* = 32 (13.7%)	*n* = 61 (26.1%)	*n* = 141 (60.1%)	
WBC (10^9^/L)	7.44 ± 2.66	8.12 ± 2.75	7.66 ± 2.44	0.424
Hb (g/dL)	13.49 ± 1.25	13.51 ± 0.87	13.55 ± 1.10	0.947
PLT (10^9^/L)	310.35 ± 71.40	311.65 ± 83.74	304.53 ± 64.25	0.798
ALT (IU/L)	16.37 ± 9.35	21.15 ± 19.76	19.7 ± 39.88	0.812
BUN (mg/dL)	11.75 ± 2.89	11.7 ± 3.28	12.17 ± 3.53	0.642
Cre (mg/dL)	0.64 ± 0.21	0.56 ± 0.20	0.6 ± 0.24	0.243
Na (mEq/L)	140.27 ± 2.36	139.72 ± 1.65	140.26 ± 1.86	0.202
K (mEq/L)	4.1 ± 0.28	4.13 ± 0.28	4.18 ± 0.28	0.27

Values are presented as mean ± standard deviation. AST, aspartate aminotransferase; ALT, alanine aminotransferase; BUN, blood urea nitrogen; Cre, creatinine; Glu, glucose; Na, sodium; K, potassium; WBC, white blood cell; Hb, hemoglobin; PLT, platelet.

**Table 3 children-09-01241-t003:** Comparison of imaging findings, and whether patients attended the emergency department or not between groups.

		AR Group	Non-AR Group	Headache Group	*p*-Value
		(*n* = 32)	(*n* = 61)	(*n* = 141)	
Emergency visit	Yes	8 (16.3%)	16 (32.7%)	25 (51%)	0.303
	No	24 (13%)	45 (24.3%)	116 (62.7%)	
Imaging Findings	Normal	18 (11.3%)	35 (21.9%)	107 (66.9%)	0.010 *
	Sinusitis, Mastoditis	14 (18.9%)	26 (35.1%)	34 (45.9%)	

* Statistically significant.

**Table 4 children-09-01241-t004:** Comparison of headache severity.

NRS	Mild	Moderate	Severe	*p*-Value
	*n* = 56	*n* = 131	*n* = 47	
BMI	18.96 ± 3.62	21.35 ± 4.05	20.01 ± 3.35	˂0.001 *
Emergency visit	7 (14.3%)	32 (65.3%)	10 (20.4%)	0.183
Abnormal Imaging Findings	17 (23%)	41 (55.4%)	16 (21.6%)	0.927
AR Group	8 (25%)	21 (65.6%)	3 (9.4%)	0.522
Headache Group	14 (23%)	32 (52.5%)	15 (24.6%)	
Non-AR Group	34 (24.1%)	78 (55.3%)	29 (20.6%)	

BMI, Body Mass Index; NRS, numerical rating scale; * Statistically significant.

## Data Availability

The data supporting the findings of this study are private due to the protection of personal data.

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
