# Peer review of "A Comparison of Clinical Features of Youth with and without Rhinitis Signs and Symptoms Who Are Hospitalized for Headache"

_children, 2022, doi:10.3390/children9081241_

Round 1

Reviewer 1 Report

Dear authors, 
The following are some suggestions and requests for clarification to make the study clearer and understandable: 

Please clarify, starting from the title, whether you want to investigate only the allergy or also other factors affecting the development and maintenance of chronic headache. 
The role of allergy is an interesting topic but needs to be better clarified in the various parts of the paper. 

-The introduction is not well deepened; it is not clear whether you would analyze the allergy factor in general or only allergic rhinitis.  The studies on this aspect (allergy in general or allergic rhinitis) are not extensively elucidated so it is not inferred what is already known about the topic and what contribution your study would like to make. Moreover, the sinusitis factor, then reported among the results, is not mentioned in the introduction. 
-Regarding materials and methods:
you may first define the type of headache to be investigated, whether primary, secondary or both, how the patients were selected for headache diagnosis, and what are the criteria for defining it as chronic. Also please specify whether it is sufficient to have performed at least one or more of the indicated instrumented examinations. 
It is also unclear whether patients were evaluated only for rhinitis or also for other allergic manifestations, and if all patients had allergy testing (SPT or inhalant multiple allergen simultaneous test). 
It should also be specified whether allergic rhinitis was diagnosed and classified by a specific score (nasal symptom score), and it would be interesting to correlate it with headache severity. 
Exclusion of individuals under 5 years of age cannot be based on the low probability of having or not having allergy. In fact, diagnosis of allergy can occur at all ages. 
In the flow chart (fig 1) I suggest changing the name of the third group (headache group) by better specifying that these are patients only with headache without rhinitis (for example “headache only group”). It is also not clear where the caption in fig 1 ends and where the text continues. 
Please also specify the reason for performing the reported laboratory tests and what is expected to be found altered in the three groups. 
-About the results: 
I suggest starting with a description of the demographics of the patients and then the clinical data (the description of the three groups, the presence of sinusitis, and the laboratory results). 
Specify how the diagnosis of sinusitis was made (e.g., clinical and instrumental data, instrumental only, ENT evaluation).
Specify what you want to investigate with instrumental examinations (presence of sinusitis or also other encephalic changes?), report in the text as well as in the table the type of abnormal finding and describe in which group is the imaging abnormality most noticeable.
-In the discussion: 
The date of "39.8% of patients with headache and allergy or AR" at line 174 is new, not reported in the results, unclear, and does not match the original division into three groups. 
Finally please discuss and compare more deeply your results on the finding of allergy, allergic rhinitis and sinusitis in patients with headache with what is reported in the literature. Also provide data from other works not only on migraine but also on other types of headaches. 
Thank you 

Author Response

Thank you for all the valuable recommendations and insights, which we believe have enriched the manuscript and give us more balanced perspective of our research. The manuscript has been carefully rechecked, and the necessary changes have been made according to your suggestions

Reviewer 2 Report

Please add how the normal distribution of data was checked. The authors have reported data with mean and standard deviation and applied ANOVA that are related to parametric statistics, so it is assumed that the data have been normally distributed, but please add the test used and clarify this point in the revised version of the manuscript. 

Please add the Y-axis title and units in the figure of VAS (Fig 4). Is that VAS 100 mm? 

Please add the Y-axis for Fig 2 and 3 as well, are those numbers of cases? 

Please add the limitations of this study. Retrospective studies have some general limitations, but please add those, in addition to the specific limitations of this study. 

In the conclusion, please first conclude the results of this study, i.e., there was no correlation. It is read in the first lines that there still might be the case although the results of this study did not show it. It can be misleading. Please revise, and consider that conclusions are generally what is concluded from the current study, not past or future studies. The assumptions can be added at the end of the discussion, as the authors also mentioned other studies that have found some correlations. Please also add a bit more in the drug-related part of the discussion as to how the application of correlational findings can help in the prevention or treatment of headaches in children. 

Did the author look into sex differences (differences between girls and boys)? The average age is 11 years in this study and it is assumed that it is before puberty? so that perhaps there has not been a statistically significant difference? but please clarify about this point and add. 

Author Response

(The authors gave the same response as above.)

Round 2

Reviewer 1 Report

Thank you for your comprehensive response to my comments. Title is the only aspect which remains unclear and that I suggest you change. 

You could modify it for example as follows: " Analysis of clinical features and factors influencing the development of pediatric headaches such as allergic rhinitis" or use the first sentence of the abstract by writing "analysis of the clinical features of pediatric headaches and the effects of allergic rhinitis on chronic headaches".

Author Response

Reviewer

You could modify it for example as follows: " Analysis of clinical features and factors influencing the development of pediatric headaches such as allergic rhinitis" or use the first sentence of the abstract by writing "analysis of the clinical features of pediatric headaches and the effects of allergic rhinitis on chronic headaches".

Answer

I appreciate you reviewing our manuscript and recommending an appropriated title. The titles that you suggested were considered suitable for our study so we modified our title as follow:

Analysis of clinical features and factors influencing the development of pediatric headaches such as allergic rhinitis